# Ultra-High Early Strength Cementitious Grout Suitable for Additive Manufacturing Applications Fabricated by Using Graphene Oxide and Viscosity Modifying Agents

**DOI:** 10.3390/polym12122900

**Published:** 2020-12-03

**Authors:** Alyaa Mohammed, Nihad Tareq Khshain Al-Saadi

**Affiliations:** School of Engineering and Mathematical Sciences, La Trobe University, Melbourne, VIC 3086, Australia; N.Al-Saadi@latrobe.edu.au

**Keywords:** cement, concrete, grout, graphene oxide, viscosity modifying agents, additive manufacturing, 3D printing

## Abstract

One of the considerable challenges in the design of cementitious mixtures for additive manufacturing/three-dimensional (3D) printing applications is achieving both suitable fresh properties and significant mechanical strengths. This paper presents the use of graphene oxide (GO) as a promising nano reinforcement material with the potential to improve the printing feasibility and quality of a 3D printed cementitious matrix. Additionally, in this study, a viscosity modifying agent (VMA) was employed as a chemical additive to attain the required consistency and flow. The printed mixture was fabricated using various cementitious materials and waste materials. This study investigated the impact of GO and VMA on the enhancement of the 3D printing of cementitious composites through several tests. A flow test was conducted using the flow table test. The results showed a high fluidity and practical consistency, which are essential for nozzle pumping and accurateness in printed shapes. Furthermore, the bleeding test showed minimal bleeding up to hardening, and a considerable self-cleaning ability was noted during handling when conducting examinations of fresh properties. For hardened properties, the mechanical strengths were exceptionally high, especially at early ages, which is crucial for the stability of sequence layers of printed composites. The tensile strengths were 3.77, 10.5, 13.35, and 18.83 MPa at 1, 3, 7, and 28 days, respectively, and the compressive strengths were 25.1, 68.4, 85.6, and 125.4 MPa at 1, 3, 7, and 28 days, respectively. The test results showed the effectiveness of the fabricated cementitious mixture design method for meeting the requirements for 3D concrete printing applications.

## 1. Introduction

Since its discovery in 2006, graphene and graphene-based materials have attracted considerable attention in the scientific literature. Graphene-based materials have applications in a broad disciplinary field, from engineering and biomedicine to aerospace. In construction engineering, graphene-based materials are used to enhance the performance of concrete and cementitious composites. Several studies have reported that using graphene-based materials improved the hydration of cement and resulted in an enhancement of the overall performance of the cementitious matrix [1,2]. Graphene oxide (GO) is one of the well-known graphene precursors, and is used as an additive in producing cementitious composites. Studies have indicated that GO sheets play an essential role in improving the mechanical strength and durability of cementitious composites [3,4]. They can increase the surface area of contact between cement particles and water, improving the hydration process, and decrease the amount of water required to wet cement particles [4]. Therefore, GO appears to be an attractive element for use as a cementitious mixture for three-dimensional (3D) printing.

Furthermore, a viscosity modifying agent (VMA) has a functional role in altering the rheological properties of cementitious mixtures. The rheological properties of fresh cementitious mixtures can be described in terms of the plastic viscosity and yield value. The plastic viscosity of a cementitious mixture can be described as the resistance to flow under external stress, while the yield stress is connected to the vital force required to cause a flow of cementitious mixture [5]. To effectively exploit the functional/mechanical properties of 3D printed concrete products, the layered extrusion of concrete elements requires the freshly printed material to have some specific rheological properties [6,7]. First is the pumping ability of batched materials, which is required to drive movement without congestion towards the printing head in a practical way. Second is the extrudability of batched material from the printing head. It is worth noting that the flow of batched materials is affected by the cohesiveness and viscosity of batched material. Last is the buildability of printed materials, including their ability to remain in stable shapes and layers, without bucking and deformation. Therefore, VMA can modify the consistency and flow of the cementitious mixture to address the requirements of 3D printing criteria.

It is true to say that the traditional way of using concrete in the construction sector on site has been significantly replaced in the last two decades through pre-casting and prefabrication. Recently, outstanding technology additive manufacturing in the construction sector has attracted considerable attention as a promising path for revolutionizing construction engineering [8,9]. The additive manufacturing technique is a 3D printing system which can fabricate almost any 3D object with a complex shape using digital data. The additive manufacturing technique specifies the process of making a solid structure by adding material. It can make a 3D structure with a specific geometry by using a 3D computer-aided design (CAD) or 3D object-scanner via computer control. In this technique, the materials are placed, combined, and hardened together layer by layer, one over the other [10]. Today, the additive manufacturing technique has moved from prototyping to producing comprehensively diverse materials in different industrial applications, such as automotive, medical, biomedical, energy, architectural, aerospace, furniture, and jewelry industries. This is mostly due to the outstanding advantages of the additive manufacturing system, which include, but are not limited to, mass customization; less or no waste; the cost effectiveness; its environmentally friendly natures; the flexibility in design; and its immediate applicability, especially for small and medium quantities of customized products [11,12,13].

The combination of cementitious materials and digital fabrication techniques has led to the development of innovative manufacturing processes for fabricating concrete-like products [14,15,16]. However, using this technology in construction engineering can be considered as being in its early stages. Several vital challenges prevent it from being extensively implemented in the construction industry. The primary and most severe challenge is the high costs involved when getting started. Therefore, the majority of construction work is conducted using a 3D printing technique based on laboratory settings. Reducing the total costs is beneficial for the long-term work-life by lowering the tooling and laborer costs for large-scale 3D printed concrete. In this regard, the next technological development must focus on additive manufacturing. Another challenge is the lack of available resource materials. The primary use of Portland cement is in producing concrete for the construction industry. However, the second primary use of Portland cement is as a cementing agent in various applications, such as grouting, soil stabilization, and backfilling. These applications of Portland cement have created a high demand for it in civil engineering projects. At the same time, there is an increasing shortage of raw materials for cement production associated with the necessity to reduce greenhouse gas emissions from cement plants. Therefore, it is essential to minimize reliance on Portland cement in civil engineering applications. Using the additive manufacturing system could be an effective way of reducing the use of Portland cement by facilitating the use of other cementitious materials for construction [17,18]. In addition to these challenges, large-scale production is still a challenge in additive manufacturing for construction applications due to the weak mechanical properties and anisotropic behavior of the printed objects and structures. The cementitious material should develop and provide high enough early strengths to support the following printed layers without buckling [16,19]. Using nanomaterials and chemical additives is a promising path. Therefore, in this study, GO and VMA were used to manipulate the properties of the printed mixture in the right way and to develop functional, printable, and ultra-high early strength cementitious grout suitable for 3D printing concrete.

## 2. Research Significance

This study presents an innovative composite of cementitious materials—a fabricated printed cementitious mixture (FPCM)—suitable for 3D printing concrete applications. FPCM has suitable handling features for enabling feasible printing and an ultra-high early strength for ensuring the buildability of the printed layers. Moreover, FPCM has a considerable self-cleaning ability, which highlights a promising path for eliminating blockage in the machinery parts. An adequate mixture design was adopted using GO and VMA to achieve significance in the performance of FPCM.

## 3. Materials and Methods

### 3.1. Materials

The fabricated printed cementitious mixture (FPCM) was made using ordinary Portland cement, industrial waste materials, and chemical additives. Table 1 shows the properties of cement. A range of industrial waste materials was used in the fabrication of FPCM to work as fillers and to increase the bulk density of the mixture. Chemical additives were mostly GO and VMA. The GO solution was supplied by Graphene Supermarket, the USA, and had a concentration of 500 mg/L and flake size of 0.3–0.7 microns. Table 2 shows the chemical composition of GO. Figure 1 presents the X-ray diffraction (XRD) analysis of GO. XRD analysis demonstrates an intense peak at 10°. When a natural graphite flake is wholly oxidized to GO, a new significant broad peak at 10° appears due to the increase in spacing between the basal planes from 0.34 to about 0.7 nm [20].

### 3.2. Fabrication of Printable Cementitious Grout

A mechanical mixer was used to mix components of FPCM according to the BS EN 196-1 [21], BS EN 196-3 [22], and BS EN 480-1 [23] specifications. Table 3 shows the fractions of components of FPCM. Firstly, the dry materials were placed in the mixer bowl (5 L capacity) and mixed at a slow speed (62 rpm). Then, water and other chemical additives were added in batches. The mixing rate was raised to 280 rpm to maintain good mixing of all ingredients. After that, the specimens were air-cured for 24 h in their molds. Finally, all specimens were demolded and moist-cured at room temperature (temperature of 20 °C and relative humidity > 95%) for an additional period of time to conduct strength tests.

On the other hand, a traditional cementitious mix (TM) was fabricated for comparison purposes for investigating the stress–strain behavior. TM consisted of General Purpose Cement (Type GP), manufactured by Cement Australia Pty Ltd., Queensland, Australia and sand (fineness modulus of 2.5), cement to sand (1:3), and a ratio of water to cement of 0.45.

## 4. Experimental Tests

Several standard tests were conducted to characterize the FPCM mixture. Tests of the flow, setting times, bleeding, permeability, compressive strength, tensile strength, and bond strength were conducted for FPCM. Lastly, the printing quality of FPCM was represented by the shape stability and printability.

### 4.1. Flow Table Test

A flow table test was conducted based on ASTM C1437 [24], in order to evaluate the workability of the cementitious mixture. A fabricated mini cone (42 × 20 × 60 mm^3^) was used in this test. At the start of the test, the cone was placed on the flow table and filled with mortar without external compaction. Then, after 30 s, the cone was lifted, and the FPCM mixture was allowed to flow freely. The initial base diameter was measured at this time, and the final diameter was measured after the flow table was drooped 25 times in 15 s. The flow value was calculated as a percentage based on the increase in the average base diameter compared to the initial base diameter. Figure 2 shows the flow test of cementitious composites.

### 4.2. Initial and Final Setting Times Test

The initial and final setting times for FPCM were determined based on ASTM C191 [25]. Vicat apparatus (Vtech Materials Tests, Victoria, Australia) was used in this test. The cementitious specimens were tested with a measurement interval of 5 min at a 23 °C temperature and 60% relative humidity. During the test measurements, a glass plate was used to cover the cementitious specimens to reduce evaporation from the specimens’ surface. Figure 3 shows the setting time test of cementitious composites using the Vicat Needle.

### 4.3. Bleeding Test

The bleeding test is an important test that can reflect the homogeneity of the cementitious mixture and its resistance to segregation. Therefore, it was used in this study to examine the stability of fresh stage FPCM. The test was conducted following ASTM C232 [26], in laboratory conditions, where the temperature was 21 °C and relative humidity was 64%.

A container with an 8 L capacity was used in the bleeding test. FPCM specimens were covered with a plastic sheet to prevent and reduce evaporation of the bleed water during the test. The bleed water was collected using a pipette at 5 min intervals until no water was observed on the surface of the test specimen. The accumulated bleed water, expressed as a percentage of the net mixing water, can be calculated as follows:Bleeding (%) = (*D*/*C*) × 100,(1)
where *D* = the accumulated mass of bleed water, (g), and *C* = the mass of net mixing water, (g).

### 4.4. Permeability Test

The examination of air permeability for FPCM was conducted following a similar test of ASTM D737 [27], by using a Proceq Torrent permeability tester, as shown in Figure 4. The specimen size was 300 × 300 × 25 mm^3^. This method is based on applying airflow at the right angles on the surface of the test specimen. Then, the concrete or cement permeability coefficient kT can be calculated using a simple theoretical model.

### 4.5. Mechanical Strengths Test

The compressive and splitting tensile strengths of the cementitious mixtures (FPCM) were obtained based on ASTM C109 [28] and ASTM C496 [29], respectively. fifty millimeter cubic specimens were used for measuring the compressive strength and cylindrical specimens of a 50 mm diameter and 100 mm height were used for measuring the splitting tensile strength of cementitious composites. The compressive and tensile tests were conducted using a universal testing machine (MTS). Three test specimens were examined for each testing period.

### 4.6. Bond Strength Test

The bond between the printed layers is another critical factor in any additive manufacturing system. Therefore, to ensure that there is adequate bonding and efficient load transfer between the printed layers of the cementitious composite, a bond strength test is required. In this study, a pull-off test was used to examine the bonding capacity of FPCM. The test was conducted based on ASTM C1583 [30]. In the test, a 5 mm layer of FPCM was placed on a concrete surface and then covered with a plastic sheet for 28 days of curing. During the test, a 50 mm diameter dummy (loading fixture) was attached to each test specimen using an adhesive agent (Epoxy MBrace). Then, the DYNA Z16 apparatus (Proceq SA, Schwerzenbach, Switzerland) was employed to perform the tests using a loading rate of 0.1 MPa/s, as shown in Figure 5.

## 5. Results

Table 4 shows the results of the fresh property tests. Figure 6 presents the development of the mechanical strengths of FPCM over time. The bond strength of FPCM specimens was 1.68 MPa. Figure 7 shows the stress–strain curve of FPCM compared with a traditional cementitious mix (TM). All of the results of the tests are the mean of three tested specimens. A more in-depth explanation of the results of the tests is given in the discussion section.

## 6. Discussion

### 6.1. Fresh Properties and Permeability Test

FPCM showed a flow of 5.5% (Table 4) and an initial liquid-like structure with a moderate viscosity. The low flow value of FPCM could be attributed to both GO and VMA. Some researchers have referred to the reduction effect of GO on the workability of cementitious materials [31,32]. Due to the high surface area of GO, most of the water is used to wet the surfaces of GO sheets. As a result, a limited amount of water could participate in enhancing the workability. Furthermore, VMA has a reduction effect on the workability of cementitious matrixes [33]. In the case of a large batch, the addition of superplasticizer is required to mitigate the low workability. However, such a reduction in the flow of FPCM did not affect its high self-compacting property and 3D printing property. Non-homogeneity or segregation was not observed during the test; this indicates an adequate cohesion stability of FPCM. Furthermore, the self-consolidation ability of FPCM could assist in avoiding the formation of air voids during the building of the printed cementitious mixture layer by layer.

The test results for the initial and final setting times of FPCM demonstrated a desirable practice period (Table 3). The initial setting occurred at 90 min, compared to 120 min for traditional cementitious mixes. This effect is possibly due to VMA addition and the high surface area of GO, as discussed earlier. The same trend was observed for the final setting time, with a value of 380 min compared to 420 min for traditional cementitious mixes. The period between the printed layers should be long enough to develop the required early mechanical strengths with a good bond strength [34].

The bleeding of FPCM specimens was minimal, with a value of about 0.1% (Table 3) and a very minimal bleeding time of about 20 min, as shown in Figure 8. As can be seen from Figure 8, FPCM displayed a neglectable bleeding tendency, even after 20 min of fabricating time. The low value of bleeding demonstrates GO’s high ability to maintain water molecules inside the cementitious mixture due to strong hydrogen bonds. Consequently, the role of GO in restricting the migration of water from interior parts of the cementitious matrix towards the surface can be explained through the influence of GO to interlock H_2_O molecules.

Furthermore, VMA also plays an important role in controlling bleeding. Long-chain molecules in VMA adhere to the surfaces of water molecules and hinder their migration to the surface, increasing the viscosity and yield stress of cementitious mixture [35].

The air permeability of FPCM was as low as 0.025 × 10^−16^ m^2^ (Table 4). The test was conducted using a Proceq Torrent permeability tester (Figure 4). This method measures the intrinsic permeability of the test specimen. The intrinsic permeability is only reliant on the internal properties of the cementitious matrix; the properties of the migrating gas or fluid have no impact on it. In this case, the direct measurement of the permeability of concrete or cement composites follows Darcy’s law and steady-state flow conditions. The air permeability test is a good measure and index of the characterization of concrete and cement [36,37]. GO and VMA caused a reduction in the permeability of FPCM. Sheets of GO can refine the microstructure of the cementitious matrix. This results in a denser cementitious structure, making it hard for the migrating fluids to transport within such a matrix [4]. This result is in good agreement with the findings of the bleeding test, where a minimal number of water molecules were able to be transported out of the cementitious matrix, upwards to the surface of the tested specimen. A similar effect was reported for VMA activity in the cementitious matrix. VMA owes its functional behavior to its molecular structure; it has high molecular weight polymers in this structure with a high affinity to water. The functional groups of the polymer molecules interact with the water and the surfaces of the fine materials. Then, VMA forms a three-dimensional structure in the liquid phase of the matrix. Such a system increases the viscosity of the matrix and internal friction. At the same time, these three-dimensional suspensions work like physical obstacles that limit the transport of migrating fluids.

### 6.2. Mechanical Strengths and Bond Strength

The cementitious composite has superior mechanical strengths, especially at early ages, as shown in Figure 6. An early strength is vital for 3D printing concrete applications as the required cementitious composite should have sufficient mechanical strengths at earlier ages, with adequate yielding stress to support a load of subsequent layers [19].

These results are compatible with the results of fresh properties. As discussed earlier, FPCM fulfilled a wide range of fresh state requirements. As a result, significantly enhanced strengths were achieved for FPCM. By properly engineering FPCM, a homogeneous mixture can be maintained during handling and the effects of bleeding and segregation can be avoided. These effects, if they exist, can result in a reduction of the quality and durability of the hardened cementitious matrix. The increase in compressive strength with age is significant; FPCM has a 1-day compressive strength of 25.1 MPa and 3.77 MPa tensile strength. Following the trend in developing ultra-high mechanical strengths, FPCM has a 125.4 MPa compressive strength and 18.83 MPa tensile strength at 28 days.

Additionally, the stress–strain curve of FPCM shifted towards higher stress values compared to a traditional cementitious mix (TM), as can be seen in Figure 7. This result indicates an increase in failure strain due to GO addition in the cement matrix, which indicates an improvement in the deformation ability of cementitious material [38,39]. Liu et al. [40], and Musso et al. [41] reported a similar result, where treated carbon nanotubes (CNTs) were used to fabricate the cement mix. They stated that the addition of the treated CNTs increased the compressive strength and failure strain of the cement mix.

Lavagna et al. [42] reported that oxide carbon fibers improved the mechanical strengths and electrical conductivity. However, the increase of the compressive strength was not as great as the increase in the compressive strength due to the addition of GO. In addition, Chaipanich et al. [43] reported a similar compressive strength for a cement mix with CNTs and fly ash compared to the control mix. It can be concluded that the GO content significantly contributed to increasing the tensile strength. The is mainly because of the good dispersion of GO in the cementitious matrix, which led to the optimum utilization of the GO flake strength.

Figure 9 and Figure 10 show the compressive and tensile tests of FPCM specimens, respectively. These figures display the crack pattern of the tested specimens. It is hypothesized that microcracking in FPCM specimens at the entire cross-section was initiated before the maximum load was reached. At this stage, GO layers play an important role in resisting the propagation of microcracks in the cementitious composites, making them hard and increasing the time required for microcracks to unify in one macrocrack. As a result, a high value of the peak load could be achieved, with considerable signs of ductile behavior. After reaching the maximum load, microcracks start to localize in one or more macrocracks, as can be seen in Figure 9 and Figure 10. It can be observed that as the mechanical strength of FPCM increases, less microcracking occurs outside the main crack path. This is in accordance with the remaining tested specimens; most of the tested specimens maintained their initial structure, without totally crashing or collapsing, by the testing machine. Spalling at the outer edges of the tested specimens was only observed in the compressive strength tests. It can be concluded that GO sheets work to bridge microcracks and prevent the crushing of tested specimens at the end of the test.

FPCM specimens presented a value of 1.68 MPa for the bond strength. This result is in good agreement with the results obtained from the mechanical strength tests. Figure 11 shows the failure mode of the pull-off tested specimens. The failure occurred in the concrete substrate, which indicates a very good bond between the FPCM and the concrete surface.

The bonding strength is particularly important in 3D printing concrete for maintaining the buildability of consequent layers. The bond strength, along with high compressive and tensile strengths, can maintain the buildability and stability of the printed cementitious matrix. GO plays an essential role in developing the bond strength of FPCM, which results from the high tensile strength of FPCM. The average modulus of GO paper determined by tensile test measurements was 32 GPa [44]. Such a high modulus of elasticity of GO sheets could effectively contribute to increasing the tensile and bonding strength of FPCM.

### 6.3. Scanning Electron Microsopy Analyses

Figure 12 shows a scanning electron microscopy (SEM) image of self-aligned GO paper, fabricated by the simple self-assembly of GO sheets by casting a 4 and 0.5 mg/mL dispersion of GO, respectively, and drying it in ambient conditions.

It has been reported that GO dispersion can form a nematic liquid-crystalline phase in an aqueous medium by the production of a high number of GO sheets [44,45]. Therefore, the dispersion liquid should have a low viscosity to maintain easy movement and accommodation of the liquid crystalline components. However, at a very low GO concentration, GO sheet dispersion can be very random, without any order. Such dispersion properties are useful for fabricating self-aligned paper structures from GO sheets [46].

The GO membrane in Figure 12a was formed by stacking several layers of GO sheets, and it was about 0.5 mm thick. In Figure 12b, the GO membrane was very thin, due to stacking to the deposit surface. There were many wrinkles, and the membrane was stressed due to losing water; as a result, many sharp edges were produced by graphene sheets [40,47]. In highly diluted GO dispersion, GO sheets can be optically observed, which can help to study the GO membrane morphology efficiently (Li et al. [48]). It can be concluded that the GO membrane stress phenomenon can effectively enhance cementitious composites’ fracture toughness and hardness by increasing the stress contact points on GO sheets. As discussed earlier, GO sheets assist in improving the mechanical strengths and bond strength and enhancing the stress–strain relationship of FPCM.

A GO sheet is usually pale, making the direct visualization of GO complicated, mostly if GO is deposited in adsorbing substrate media [46]. Therefore, it was challenging to optically identify GO sheet dispersion within the cement matrix.

For that reason, a simple mixture of the cementitious matrix (cement:sand of 1:3) was prepared with GO addition to investigate the effect of GO on hydration products.

Figure 13a shows SEM scans for surfaces from the cementitious matrix with GO. It can be seen from Figure 13a that there is a layer-like structure with some tops on it. It is believed that this is a layer of stacked GO sheets forming GO paper; it is similar to the GO prepared paper in Figure 12.

It is believed that the influence of GO on the cement microstructure depends on the GO lateral size distribution. GO has a sheet-like structure, and the lateral size of GO sheets ranged from a few nanometers to several microns, as reported by Liu et al. [40]. It is believed that GO sheets with a small size of a few nanometers act as a nucleation agent for C-S-H, which forms on the outsides of GO flacks, rather than the adjacent non-hydrated cement particles [49,50]. This is because the smaller the GO flakes, the weaker the Van der Waals forces required to collect GO sheets together [51,52], as can be seen in Figure 13b. On the other hand, GO sheets with a large lateral size of hundreds of nanometers have more vital van der Waals forces, which cause the formation of a GO paper-like structure, as shown in Figure 13a. This figure presents a GO membrane on cement hydration products.

On the other hand, small lateral size sheets can create reaction points in cement particles and produce small hydration products. These hydration products distribute uniformly within the cement matrix, as shown in Figure 13b. It can be concluded that some of the stacked GO sheets do not react with cement compounds and they form together with a paper-like structure. Tong et al. [53] stated that more C-S-H gels could be attracted around graphene nanoparticles and produce a denser microstructure, increasing the mechanical strength.

Furthermore, it can be deduced that these changes of the cement matrix microstructure induced by GO addition caused the improvements in the mechanical properties of FPCM.

### 6.4. Print Quality and Self-Cleaning

In this paper, a simplified laboratory testing method was used to investigate the printing quality of modified cementitious composite FPCM. A manual caulking gun apparatus (with a circular nozzle diameter), as shown in Figure 14a, was used to extract the cementitious mixtures and construct them as one layer. The nozzle size of the caulking gun was adjusted to 9 mm. The fresh mix of the cementitious composite was placed in the caulking gun, as shown in Figure 14b. After that, it extracted manually on a small scale, in order to simulate the last stage in the 3D printing/additive manufacturing method, as shown in Figure 14c. The printing quality was expressed by the surface quality (free of surface defects), squared edges (visible layer edges), and dimension conformity and consistency (Figure 14c). Additionally, polymer-based material such as superplasticizer and, in this study, VMA, could show the performance when employing a graphene-based material such as GO [54,55]. Based on the results of this study, it can be concluded that using VMA works well with the GO content to form a printed cementitious mixture and assist in increasing the compressive, tensile, and bond strengths. Moreover, using VMA in the mix of FPCM helps to create a mix that exhibits flowability values that allow extrudability and the ability of the extruded filament to hold its shape.

It is well-known that hardened cementitious composites require complex chemicals to clean, mostly increasing the cost of maintenance of machinery. Therefore, it is essential to emphasize proper ways to clean printing machinery; one is through using smart materials. It was noted that FPCM showed a self-cleaning feature during printing, so minimum efforts were needed to scrape the remaining cementitious mixture off the sides of equipment. It had a highly cohesive mixture and self-consolidation, so few residuals from the mix were left in the printing equipment. Lathering the remaining FPCM with fresh FPCM creates friction, which easily helps to scrub hardened cementitious mixture. Efficient cleaning, in this case, is critical to maintaining the printing machinery and reducing the cost of operating. Furthermore, using VMA could reduce the friction and pressure required to pump cementitious composites. This results in a better surface appearance and minimizes blockages in pumping lines caused by the settlement and sedimentation of ingredients in the printed materials.

## 7. Conclusions and Recommendations

Current construction technology produces highly engineered materials with adequate utilization of their characteristics; advanced construction composites could be fabricated to achieve specific needs. In this regard, an innovative mixture of FPCM was made using a range of cementitious components, GO, and VMA.

FPCM results in most of the required properties for 3D printing concrete, mostly described by suitable fresh properties and superior mechanical and bond strengths. Additionally, it is mainly produced from waste materials, which is a fundamental approach for recycling waste materials and reducing the use of Portland cement.

As a result, sustainability is significantly achieved by using this innovative mixture, as using Portland cement remarkably increases the cost of construction works.

The test results showed the effectiveness of GO in improving the printing quality of FPCM; the results confirm that the nuclear reinforcement of GO improved the output of the 3D printed cementitious matrix.

The tests of the fresh state of FPCM show an adequate workability and setting time, and a pot life of up to 90 min. These fresh properties are essential for 3D concrete printing applications.

The strength of FPCM is remarkably high, being above 125 MPa for the compression strength, 18 MPa for the tension strength, and 1.68 MPa for the bond strength. These mechanical and bond strengths are also essential for 3D concrete printing applications.

FPCM presented the self-cleaning feature during printing. The residual from the used mix was easily cleaned by lathering it with fresh FPCM; the resulting friction lifted the residuals quickly.

The test results showed the suitability of FPCM for 3D printing concrete and confirmed its potential for more development in future investigations. Therefore, it can be concluded that GO reinforcement could represent a promising path for obtaining high-quality 3D printed cementitious materials.

Recommendations for future work are as follows:The examination of other factors related to additive manufacturing systems, such as the layer height and presence of reinforcement agents;The potential application of research results in civil engineering including, but not limited to, expansive mortar, grouting, shotcrete coating, and ground anchors;The findings of this study can be utilized to design and fabricate a wide range of engineered cementitious materials that can be used for different civil and construction applications, for example, insulation material for freeze-thaw and fire-proofing material, soil stabilization, backfilling, and lining materials of very complicated sites such as underground structures.

## 8. Patents

Adhesive composition and a method of preparing an adhesive, Australian Number 2017901649, is a patent relating to the work reported in this manuscript.

## Figures and Tables

**Figure 1 polymers-12-02900-f001:**
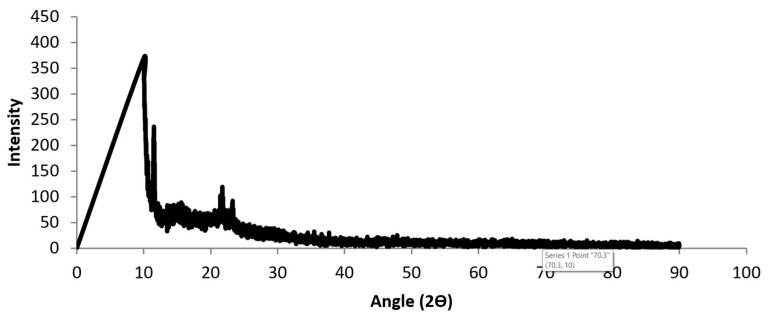
X-ray diffraction (XRD) analyses of graphene oxide (GO).

**Figure 2 polymers-12-02900-f002:**
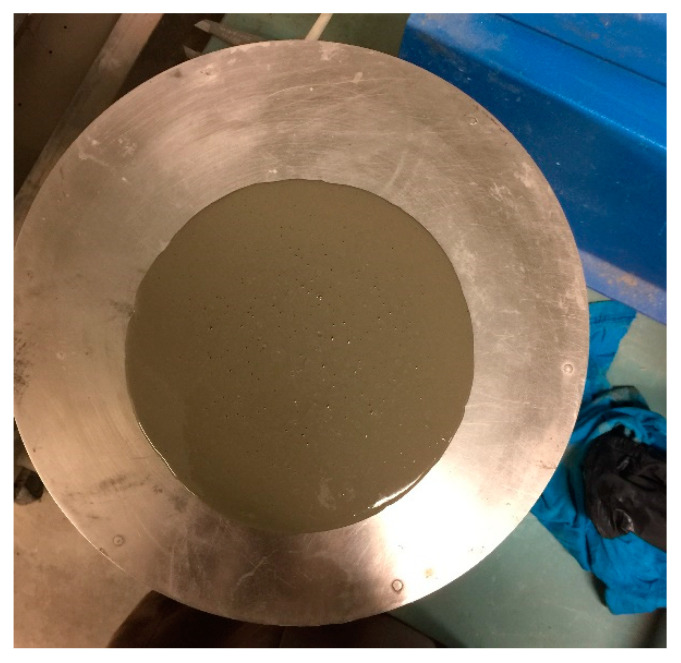
Flow table test of cementitious composites.

**Figure 3 polymers-12-02900-f003:**
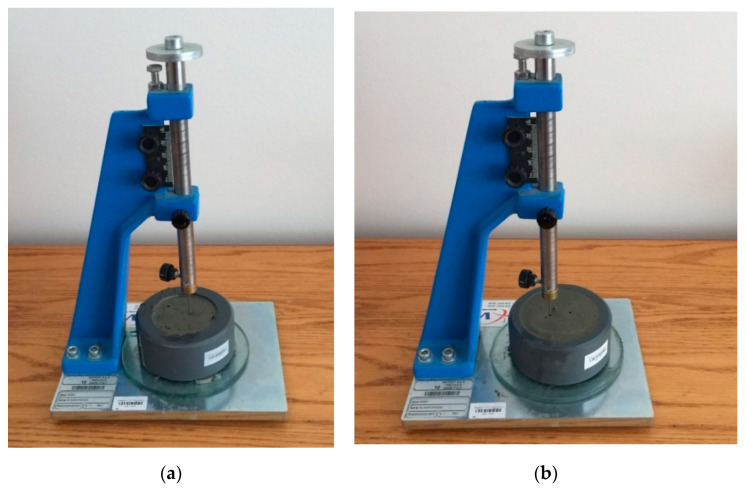
Setting time test of FPCM: (**a**) 15 min and (**b**) 30 min.

**Figure 4 polymers-12-02900-f004:**
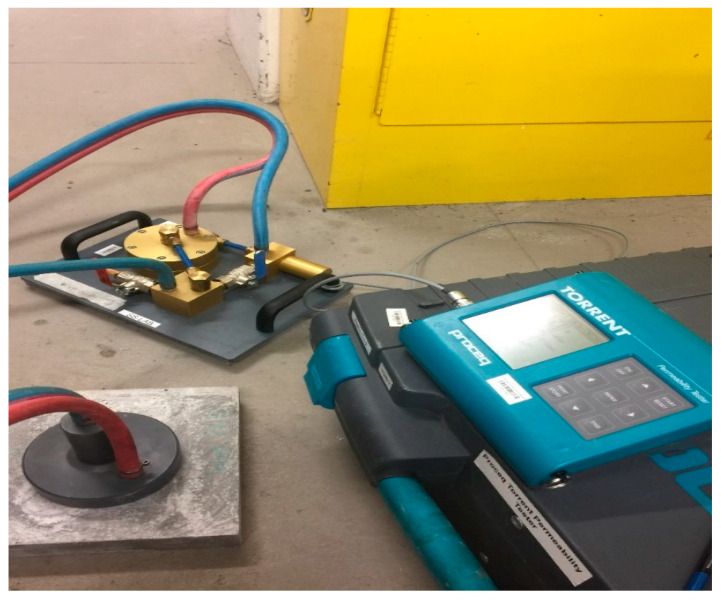
Permeability test of cementitious specimens.

**Figure 5 polymers-12-02900-f005:**
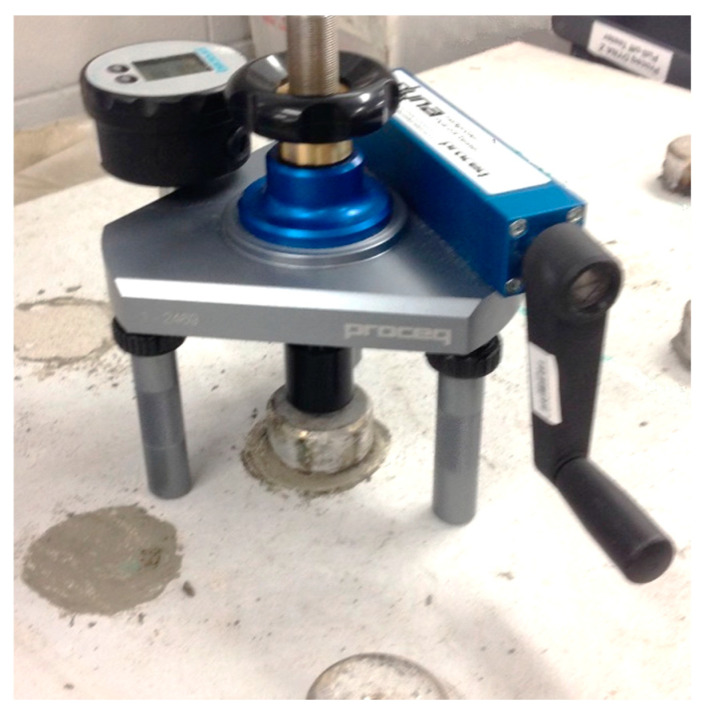
Pull-off test of cementitious specimens.

**Figure 6 polymers-12-02900-f006:**
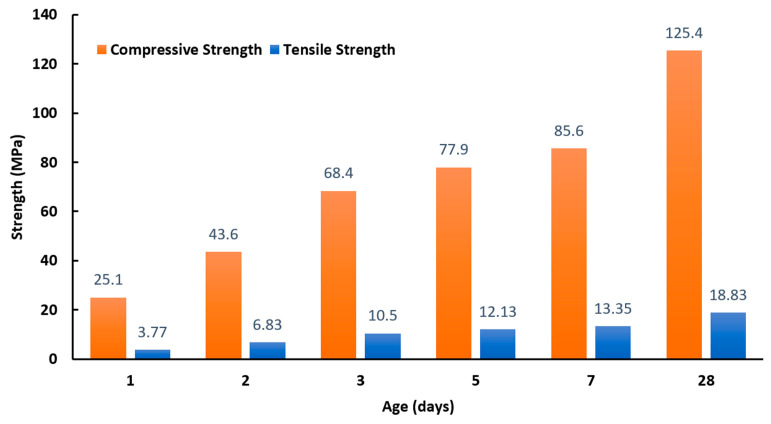
Development of mechanical strengths over time for FPCM (note: The coefficients of variation range between ±0.3% and ±1.7%).

**Figure 7 polymers-12-02900-f007:**
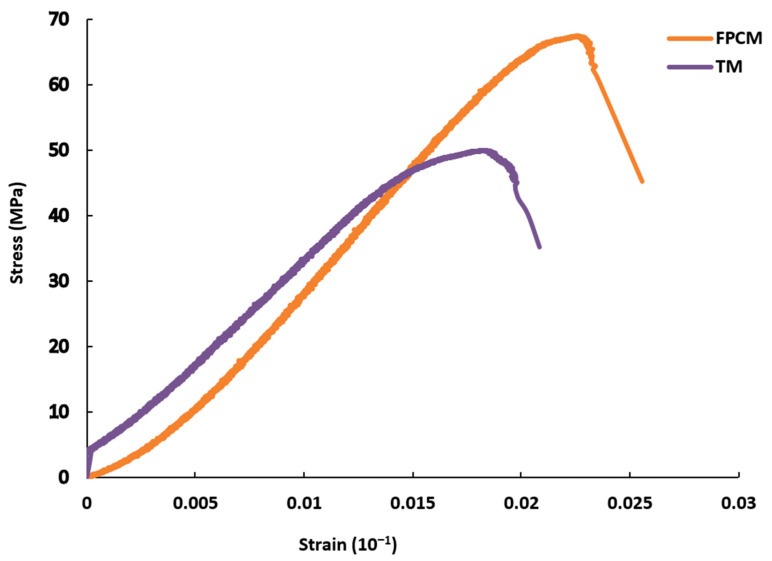
Stress–strain behavior of FPCM at the age of 3-days compared to the stress–strain of a traditional cementitious mix (TM) at the age of 28-days.

**Figure 8 polymers-12-02900-f008:**
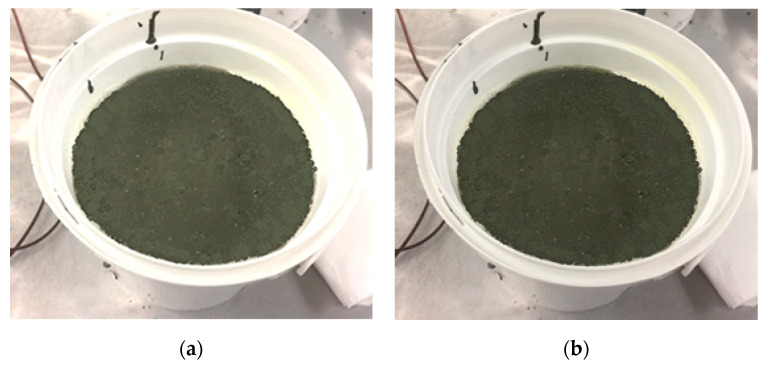
Bleeding test of FPCM (**a**) at 1 min and (**b**) at 20 min.

**Figure 9 polymers-12-02900-f009:**
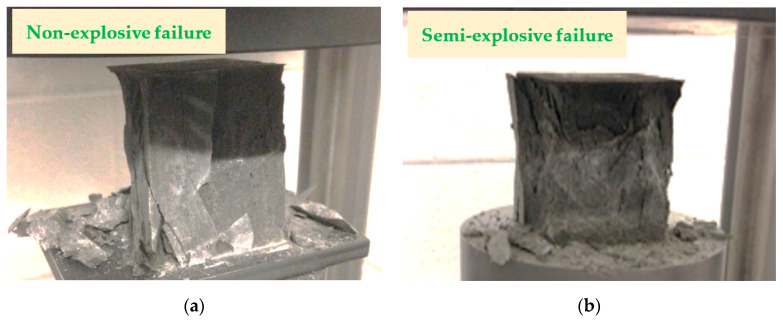
Compressive strength test of FPCM (**a**) at the age of 7 days and (**b**) at the age of 28 days.

**Figure 10 polymers-12-02900-f010:**
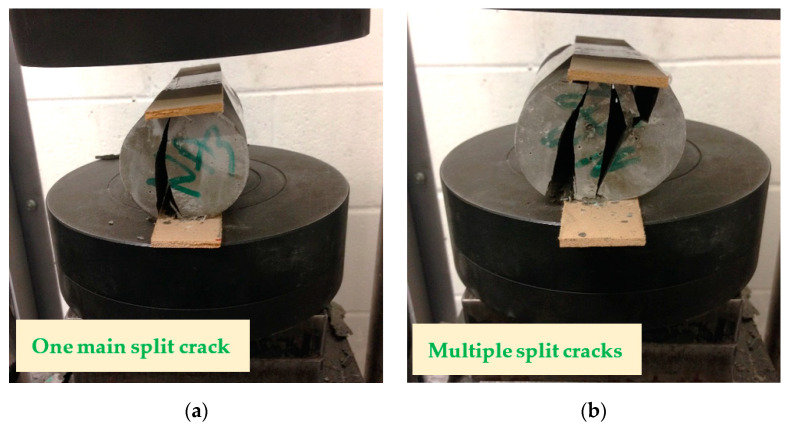
Tensile strength test of FPCM (**a**) at the age of 7 days and (**b**) at the age of 28 days.

**Figure 11 polymers-12-02900-f011:**
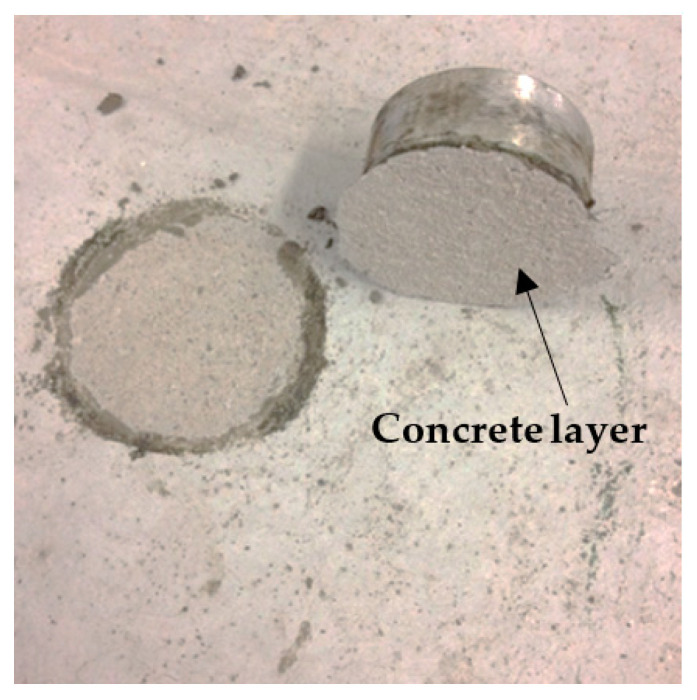
Pull-off test of FPCM in failure mode.

**Figure 12 polymers-12-02900-f012:**
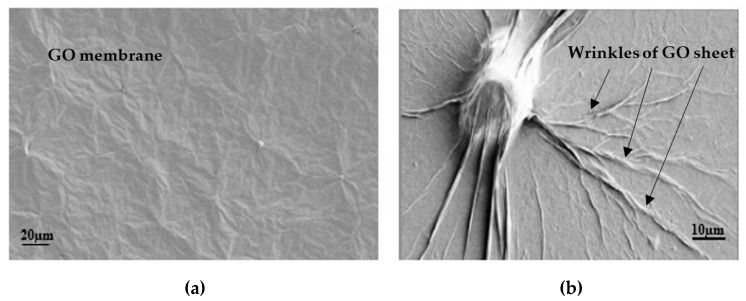
Scanning electron microscopy (SEM) image of the self-assembly of GO paper with different concentrations: (**a**) 4 mg/mL and (**b**) 0.5 mg/mL.

**Figure 13 polymers-12-02900-f013:**
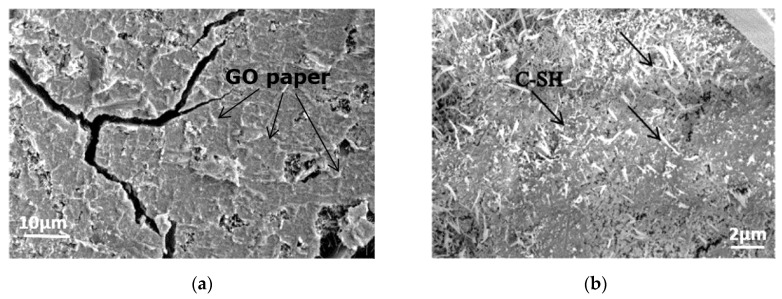
SEM images of (**a**) a GO paper-like structure on cement hydration products and (**b**) point reaction spots of small GO sheets.

**Figure 14 polymers-12-02900-f014:**
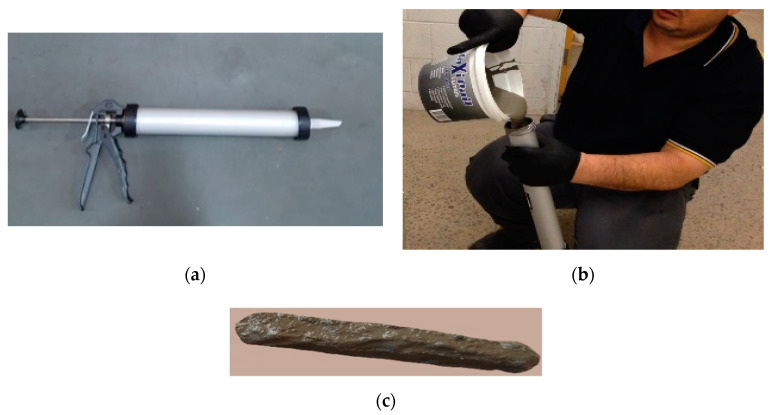
(**a**) Caulking gun apparatus with a nozzle diameter of 9 mm; (**b**) liquid-like structure of cementitious composites; and (**c**) printed coupon specimen of FPCM made by using a caulking gun.

**Table 1 polymers-12-02900-t001:** Chemical composition of cement.

Test	%
Portland Clinker	86–95
Gypsum	5–8
Sulphur Trioxide	2.4
Chloride	0.02
Equivalent Alkalis	0.5
Crystalline Silica	<1
Mineral Addition	Up to 7.5

**Table 2 polymers-12-02900-t002:** Chemical composition of graphene oxide (GO).

Element	%
Carbon	79
Hydrogen	0–1
Sulphur	0–1
Nitrogen	0–1
Oxygen	20

**Table 3 polymers-12-02900-t003:** Fractions of the composition of fabricated printed cementitious mixture (FPCM).

Component	%
Cement	30
Cementitious ingredients	16.75
Mineral additives	23
Chemical additives	2.25
Filler	28
Water/cement	14

**Table 4 polymers-12-02900-t004:** Test results on the fresh properties of FPCM.

Flow Table Test	Initial Setting Time (min)	Final Setting Time (min)	Bleeding	Segregation	Permeability × 10^−16^ m^2^
5.5%	90	380	0.1%	Non	0.025

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
