# Peer review of "Ultra-High Early Strength Cementitious Grout Suitable for Additive Manufacturing Applications Fabricated by Using Graphene Oxide and Viscosity Modifying Agents"

_polymers, 2020, doi:10.3390/polym12122900_

Round 1
Reviewer 1 Report
The article covers the topic of the Ultra-High Early Strength Cementitious Grout Suitable for 3D Printing Applications Fabricated by using Graphene Oxide and Viscosity Modifying Agents.
In my opinion, article presents valuable content. The supporting experiments are informative and present added value to the body of knowledge on the subject area. The manuscript has good cohesion.
In my opinion, the authors have achieved to provide interesting research work. This is an interesting paper that deals with a timely topic and novel idea. There are some weaknesses through the manuscript which need improvement. Some modification should be considered:
1. I suggest to add separated point 2 - Research significance - Please descibe here the main essence of the research. (Why presented
paper is so important? What is major innovation accent in presented studies?).
2. The authors have done a good job on the literature review. However, the introduction part needs more attention.
I suggest to add more literature with regards to the works that have been published in the Journal of MDPI.
The introduction part should contain deeper analysis. The authors should pay attention on the parameters, which have
influence on the flowability of concrete mixtures, ex. shape of grains, type of admixture, porosity of concrete mixture etc.
The following articles could be useful: https://doi.org/10.3390/ma11081372, https://doi.org/10.1016/j.conbuildmat.2019.117794,
3. Please present in the table the content of the concrete mixture.
4. Point 3.5 how many specimenes were tested in each case?
5. Point 3.6 - what type of glue was used in bond strength test?
6. Figure 7 - where is the TM curve? Please improve it.
7. The text: 'TM: Traditional cementitious mix: Australian manufactured OPC, Sand (fineness modulus 2.5).
Cement to sand (1:3) and the ratio of water to cement is 0.45.' should be removed from the current position and transferred
into main body of the article.
8. Figure 8 - what is the difference between the figures? Please add the scale.
9. Please improve the quality of figure 9 (b).
10. Figures: 9, 10, 11, 12 - the photographs should be described on the photographs.
11. The analysis should contain the fracture mechanism of considered specimens based on photo of specimen after test (in case
of compression and tension tests. It is also recommended to show all tested specimens.
12. The conclusion must be more than just a summary of the manuscript. I suggest that major conclusions should be presented point by point.
13. It is strongly recommended to indicate potential application of research results in civil engineering or another discipline.
14. The cited literature should be in accordance to the rules in 'Polymers'.
Author Response
Dear Editor,
Please find my response to the reviewer's comments as below:
- I suggest to add separated point 2 - Research significance - Please descibe here the main essence of the research. (Why presented
paper is so important? What is major innovation accent in presented studies?).
Point 2 was included in the revised manuscript presents Research Significance.
- The authors have done a good job on the literature review. However, the introduction part needs more attention.
I suggest to add more literature with regards to the works that have been published in the Journal of MDPI.
The introduction part should contain deeper analysis. The authors should pay attention on the parameters, which have
influence on the flowability of concrete mixtures, ex. shape of grains, type of admixture, porosity of concrete mixture etc.
The following articles could be useful: https://doi.org/10.3390/ma11081372, https://doi.org/10.1016/j.conbuildmat.2019.117794,
The authors included deep analyses of the factors mentioned by the reviewer whenever it in alignment with the study’s scope.
- Please present in the table the content of the concrete mixture.
A table (table 3) that include the fractions of the mixture was included in the revised manuscript.
- Point 3.5 how many specimens were tested in each case?
Three specimens were tested for each testing period.
- Point 3.6 - what type of glue was used in bond strength test?
Epoxy MBrace was used to fixe dummies to test specimens.
- Figure 7 - where is the TM curve? Please improve it.
Figure 7 was improved in the revised manuscript.
7. The text: 'TM: Traditional cementitious mix: Australian manufactured OPC, Sand (fineness modulus 2.5).
Cement to sand (1:3) and the ratio of water to cement is 0.45.' should be removed from the current position and transferred. into main body of the article.
These details were removed to the main body of the article in the revised manuscript in section 3.2.
8. Figure 8 - what is the difference between the figures? Please add the scale.
The difference is the time scale. Extra details included in the revised manuscript in Figure 8 for better clarity.
- Please improve the quality of figure 9 (b).
The quality of figure 9 (b) had been fixed in the revised manuscript.
- Figures: 9, 10, 11, 12 - the photographs should be described on the photographs.
The photographs had been described in the photographs for Figures: 9, 10, 11, 12 in the revised manuscript.
- The analysis should contain the fracture mechanism of considered specimens based on photo of specimen after test (in case of compression and tension tests. It is also recommended to show all tested specimens.
The fracture mechanism of the tested specimens after test are shown in Figures 9 and 10 for compression and tension tests, respectively.
12. The conclusion must be more than just a summary of the manuscript. I suggest that major conclusions should be presented point by point.
The conclusion had been presented point by point in the revised manuscript.
13. It is strongly recommended to indicate potential application of research results in civil engineering or another discipline.
The potential application of research results in civil engineering or another discipline had been added in the revised manuscript in section 7.
14. The cited literature should be in accordance to the rules in 'Polymers'.
The cited literature had been fixed in accordance to the rules in 'Polymers' as shown in the revised manuscript.
Reviewer 2 Report
The authors nicely structured the work and made a detailed analysis of all the properties that are essential for a material to be used commercially in additive manufacturing.
My comments:
1.) The title is a little too long. But at the moment I have no suggestion on how to write it differently.
2.) Both words: Additive manufacturing and 3D printing is mentioned in many places in the text (line 65, 72, 84, 91, 94). The word 3D printing should be deleted and only additive manufacturing should be written. And accordingly, the title of the paper should then be changed.
3.) Line 106: How many VMA was added in the mixture?
Have the authors considered other values of additives in the cementitious mixture?
Maybe a design of experiment needed to be made?
How did you get the concentration of 500 mg/L for GO? Is it experiential, from some literature or ....?
4.) Chapter 2.2.: How long did the mixture mix with 62 rpm and with 280 rpm?
How long did it cured at room temperature?
De moulded I think it is written together demoulded.
5.) How is the sample prepared for the bond strength test?
Is this test of bond strength of two printed layers? The test specimen was made with the caulking gun shown in Figure 13?
6.) How is the sample attached to the device in Figure 5?
7.) In the strength results, it is necessary to show the standard deviations. If authors do not want to put values from all three test specimens, then it is necessary to write a standard deviation on the diagram (figure 6) next to the mean value, eg 25.1 +/- st.dev.
8.) The curves in Figure 7 is missing. Where is the curve for TM for 3 days and for 28 days. Also FPCM is now for 3 day or 28 days? It is currently unclear what Figure 7 shows.
Why did the authors choose the 3-day and 28-day comparison? Why 3 days?
9.) Perhaps it would be clearer to put the results for TM in Table 3 as well.
10.) Line 267-268: Review the sentence. Compressive strength is mentioned twice.
11.) Put a picture of the test / result diagram for the bond strength (Line 288).
12.) The figure and the title of the figure would be better if it was on the same page. (figure 11 and 12).
Author Response
Dear Editor,
Please find my response to the reviewer's comments as below:
1.) The title is a little too long. But at the moment I have no suggestion on how to write it differently.
The title has most of the keywords of the research, and it gives a brief description of what to expect from the research. Write the title in another way may lead to interrupt the overall message that the title aims to deliver. The title of the paper had been changed following the reviewer’s suggestion, the word 3D printing was deleted, and only additive manufacturing being written in the revised manuscript.
2.) Both words: Additive manufacturing and 3D printing is mentioned in many places in the text (line 65, 72, 84, 91, 94). The word 3D printing should be deleted and only additive manufacturing should be written. And accordingly, the title of the paper should then be changed.
Following the reviewer’s suggestion, the word 3D printing was deleted, and only additive manufacturing being written in the revised manuscript.
3.) Line 106: How many VMA was added in the mixture?
VMA was added following the manufacturer’s guide, 0.1 - 1.0% by weight of cement (Sika VMA).
For the mixture, we used 0.5% VMA.
Have the authors considered other values of additives in the cementitious mixture?
Several trial mixes were made based on the manufacturer’s guide; it was found that VMA % could be adjusted to meet the required fresh properties at the time of fabricating. Also, the temperature of placing environment should be considered.
Maybe a design of experiment needed to be made?
More details about the experiment of the mixture were included in the revised manuscript in section 3.
How did you get the concentration of 500 mg/L for GO? Is it experiential, from some literature or ....?
The concentration is provided by the manufacturer’s guide.
4.) Chapter 2.2.: How long did the mixture mix with 62 rpm and with 280 rpm?
Dry ingredients mixed with 62 rpm for 1 minute and after adding liquid ingredients mixed with 280 rpm for 3 minutes.
How long did it cured at room temperature?
For the mechanical strengths, specimens were cured following a time frame for testing at ages of 1,2,3,5,7, and 28 days.
De moulded I think it is written together demoulded.
That is based on a language editor software. I checked some references, demould could be used as one word, the word was corrected in the revised manuscript.
5.) How is the sample prepared for the bond strength test?
More details about test was included in the revised manuscript.
Is this test of bond strength of two printed layers? The test specimen was made with the caulking gun shown in Figure 13?
At this study, the bond strength was tested between the mixture (FPCM) and a surface of normal concrete (a slab aged 8 months). This condition represents the most critical bond case. The performance of the mixture was significant, and it achieved a value of 1.68 MPa for bond strength. Therefore, it is expected that the mixture could achieve more than 1.68 MPa for bond strength between two fresh printed layers. These tests, between two fresh printed layers, would be examine and included in further development in the next study in the future.
6.) How is the sample attached to the device in Figure 5?
The test specimen is placed in a uniform layer attached to the concrete surface, more explanation about the pull-off test had been added in the revised manuscript.
7.) In the strength results, it is necessary to show the standard deviations. If authors do not want to put values from all three test specimens, then it is necessary to write a standard deviation on the diagram (figure 6) next to the mean value, eg 25.1 +/- st.dev.
The coefficients of variation had been added in figure 6, as shown in the revised manuscript. This Figure presents dense information, for that, the range of coefficient of variation was included in the title of the Figure,
8.) The curves in Figure 7 is missing. Where is the curve for TM for 3 days and for 28 days. Also FPCM is now for 3 day or 28 days? It is currently unclear what Figure 7 shows.
Why did the authors choose the 3-day and 28-day comparison? Why 3 days?
Figure 7 had been fixed in the revised manuscript. In Figure 7 the stress-strain relation of FPCM at 3-days was compared to the stress-strain relation of TM at 28-days. The compressive strength of FPCM at 3 days which was higher than that of TM at 28 days.
This study emphasis on early strength as an indication of buildability of mixture for additive manufacturing.
High strength at an early age such as 3 days for FPCM indicates that the ability of the initially printed layers to withstand the weight of the subsequent layers without collapse or deformation. Especially important for a large-scale printed structure where the cumulative weight of the printed layers could cause buckling in the printed structure at early ages.
9.) Perhaps it would be clearer to put the results for TM in Table 3 as well.
TM is a traditional concrete mixture, only mentioned in this study for comparison purpose.
The aim is to present the advancement of using the developed mixture over the traditional concrete.
More details were included in the revised manuscript to clear the purpose of using TM in the study.
10.) Line 267-268: Review the sentence. Compressive strength is mentioned twice.
Compressive strength is mentioned twice but in a different contest.
11.) Put a picture of the test/result diagram for the bond strength (Line 288).
The picture of the test/result diagram for the bond strength had been added in the revised manuscript.
12.) The figure and the title of the figure would be better if it was on the same page. (figure 11 and 12).
The figure and the title of the figure (figure 11 and 12) had been put on the same page, as shown in the revised manuscript.